# Role of Ubiquitination and Epigenetics in the Regulation of AhR Signaling in Carcinogenesis and Metastasis: “Albatross around the Neck” or “Blessing in Disguise”

**DOI:** 10.3390/cells12192382

**Published:** 2023-09-29

**Authors:** Ammad Ahmad Farooqi, Venera Rakhmetova, Gulnara Kapanova, Gulnur Tanbayeva, Akmaral Mussakhanova, Akmaral Abdykulova, Alma-Gul Ryskulova

**Affiliations:** 1Institute of Biomedical and Genetic Engineering (IBGE), Islamabad 54000, Pakistan; 2Department of Internal Diseases, Medical University of Astana, Astana 010000, Kazakhstan; 3Faculty of Medicine and healthcare, Al-Farabi Kazakh National University, 71 Al-Farabi Ave, Almaty 050040, Kazakhstangulnurtanbayeva@gmail.com (G.T.); 4Scientific Center of Anti-Infectious Drugs, 75 Al-Farabi Ave, Almaty 050040, Kazakhstan; 5Department of Public Health and Management, Astana Medical University, Astana 010000, Kazakhstan; makmaral1@mail.ru; 6Department of General Medical Practice, General Medicine Faculty, Asfendiyarov Kazakh National Medical University, Almaty 050000, Kazakhstan; akmaral_abd@mail.ru; 7Department of Public Health and Social Sciences, Kazakhstan Medical University “KSPH”, Utenos Str. 19A, Almaty 050060, Kazakhstan; r.alma@bk.ru

**Keywords:** cancer, apoptosis, signaling, xenografted mice, preclinical trials

## Abstract

The molecular mechanisms and signal transduction cascades evoked by the activation of aryl hydrocarbon receptor (AhR) are becoming increasingly understandable. AhR is a ligand-activated transcriptional factor that integrates environmental, dietary and metabolic cues for the pleiotropic regulation of a wide variety of mechanisms. AhR mediates transcriptional programming in a ligand-specific, context-specific and cell-type-specific manner. Pioneering cutting-edge research works have provided fascinating new insights into the mechanistic role of AhR-driven downstream signaling in a wide variety of cancers. AhR ligands derived from food, environmental contaminants and intestinal microbiota strategically activated AhR signaling and regulated multiple stages of cancer. Although AhR has classically been viewed and characterized as a ligand-regulated transcriptional factor, its role as a ubiquitin ligase is fascinating. Accordingly, recent evidence has paradigmatically shifted our understanding and urged researchers to drill down deep into these novel and clinically valuable facets of AhR biology. Our rapidly increasing realization related to AhR-mediated regulation of the ubiquitination and proteasomal degradation of different proteins has started to scratch the surface of intriguing mechanisms. Furthermore, AhR and epigenome dynamics have shown previously unprecedented complexity during multiple stages of cancer progression. AhR not only transcriptionally regulated epigenetic-associated molecules, but also worked with epigenetic-modifying enzymes during cancer progression. In this review, we have summarized the findings obtained not only from cell-culture studies, but also from animal models. Different clinical trials are currently being conducted using AhR inhibitors and PD-1 inhibitors (Pembrolizumab and nivolumab), which confirm the linchpin role of AhR-related mechanistic details in cancer progression. Therefore, further studies are required to develop a better comprehension of the many-sided and “diametrically opposed” roles of AhR in the regulation of carcinogenesis and metastatic spread of cancer cells to the secondary organs.

## 1. Introduction

Aryl hydrocarbon receptor (AhR) has a central role in crucial processes, including cellular stress responses, proliferation, inflammation and carcinogenesis [1,2,3,4]. Combined with our evolving knowledge regarding the important role of AhR, it is becoming relatively easier to understand how AhR rewires the signaling landscape. Importantly, AhR and interconnected proteome of the tumor microenvironment can drive malignant transformation, drug resistance, epithelial-to-mesenchymal transition, immune evasion and metastatic spread of cancer cells to secondary sites. Integrated experimental and computational studies have galvanized the identification of structural requirements for ligand-induced activation of AhR [5,6,7,8,9]. Different proteins are involved in nuclear accumulation of AhR [10]. Importantly, nuclear accumulation of AhR is necessary for transcriptional gene networks.

After ligand binding, AhR shuttles into the nucleus and dimerizes with ARNT (AhR nuclear translocator protein) [11,12]. Essentially, AhR/ARNT dimers transcriptionally control genetic networks, which underlie carcinogenesis and metastasis. AhR works with wide variety of co-factors and chromatin remodeling machinery (Figure 1) [13,14]. A high degree of complexity has emerged in the context of the multifaceted role of AhR in cancer, with a series of outstanding questions related to the cancer promoting and cancer suppressive activities of AhR signaling in studies. An integrated and comprehensive view of AhR functionalities in the regulation of the carcinogenesis and metastasis is expanding at a previously unprecedented momentum. A substantial fraction of mechanism-based information has deconvoluted highly intricate cellular quality-control processes. Ubiquitination of a plethora of proteins plays a linchpin role in shaping the outcome of different signaling pathways.

There are some excellent reviews ranging from description of different types of AhR-agonists to the mechanisms of AhR-mediated regulatory networks [15,16,17,18,19,20,21,22,23,24]. However, in this review we have attempted to gather some of the proof-of-concept studies related to ubiquitination, epigenetics and AhR-related molecular biology to complete missing pieces of an incomplete jigsaw puzzle. These apparently discrete mechanisms have to be viewed as a multi-dimensional perspective to comprehensively analyze the mechanisms of AhR in the inhibition/prevention of cancer.

A considerable proportion of cutting-edge seminal studies has underscored the role of AhR as a “double-edged” sword. Therefore, we have partitioned this review primarily into three sub-sections. Initially, we offer our readers a brief discussion about the oncogenic and tumor-suppressive roles of AhR specifically in the animal model studies. Later, we provide a detailed analysis of the existing mechanistic evidence related to the role of AhR in the regulation of ubiquitination as well as “moonlighting” activities of AhR as a ubiquitin ligase. The last section deals with the epigenetic aspects of AhR activities.

## 2. Oncogenic Role of AhR

AhR has been shown to promote carcinogenesis mainly through regulation of different signaling pathways. AhR-mediated rewiring of signaling pathways contextually reduced immunological responses and severely impaired natural killer cell-mediated inhibition of tumor progression.

IDO1 and TDO2 displayed the unique ability to catalyze kynurenine production, which promoted carcinogenesis by impairment of host immunosurveillance. Indoleamine 2,3-dioxygenase 1 (IDO1) catalyzed the rate-limiting steps in tryptophan catabolism and exerted immunosuppressive effects primarily through the activation of the AhR pathway. AhR activation by the IDO1/TDO2 product kynurenine promotes a tumor microenvironment that is defective in recognition and eradication of the cancer cells. Kynurenine-mediated activation of AhR leads to the immunological escape of cancer cells. AhR regulated NKG2DL/NKG2D-mediated signaling to enhance immune escape. NKG2Ds are lectin-like type-2 transmembrane stimulatory immunoreceptors. NKG2D contains a charged transmembrane residue that enables it to interact with signaling adaptor molecules. Structural studies have shown that NKG2D transduce the signals intracellularly via an Src homology 2 domain-binding site in the DAP10 adaptor. ADAM10 (a disintegrin and metalloprotease-10) played a critical role in the shedding of NKG2DL and the release of soluble NKG2DL. Importantly, AhR stimulated transcriptional activation of ADAM10 [25]. ADAM10 specific inhibitors efficiently reduced the shedding of NKG2DL and enhanced its binding to NKG2D receptors (Figure 2) [26].

As Kynurenine is an agonist of AhR, therefore AhR promotes the entry of kynurenine in the NK cells. AhR antagonized STAT proteins driven downstream signaling. Functionally active forms of STAT proteins directly bind to the promoter regions of NK receptors (NKG2D and NKp46) (Figure 2). Treatment with STAT inhibitors resulted in an evident decline in the expression of NKG2D and NKp46 [27]. It is highly relevant to mention that AhR-mediated upregulation of PD-1 and PD-L1 signaling is currently an area of intense clinical research.

A liver metastasis model was generated by the injection of TDO2-expressing CT26 cells into the spleen of Balb/c syngenic mice. There was an evidence increase in the liver metastasis in mice injected with TDO2-overexpressing CT26 cells. AhR inactivation with a specific inhibitor (CH-223191) caused a complete blockade of the TDO2-mediated increase in hepatic metastasis. AhR has been shown to transcriptionally upregulate PD-L1 and TDO2. Activation of the kynurenine-AhR cascade by TDO2 induced the activation of PD-L1 and TDO2 in colon cancer cells. Knockout of PD-L1 almost completely impaired the TDO2-induced increase in hepatic metastasis. TDO2-induced liver metastasis was also reported to be abrogated by the intraperitoneal injections of neutralizing anti-PD-L1 antibodies [28].

The tumor cells recruited the macrophages into the tumor tissues and promoted the polarization of M2 macrophages, which consequently facilitated the malignant progression of the tumor. The frequency of M2 macrophages was noted to be higher in mice subcutaneously injected with TDO2-overexpressing KYSE150 cancer cells. Tumor progression by TDO2-overexpressing KYSE150 cancer cells was abolished in macrophage depleted animal models [29]. IDO1/TDO dual inhibitor RY103 targeted the Kyn-AhR pathway in mice xenografted with pancreatic cancer cells [30].

AhR has been shown to trigger the activation of JAK/STAT signaling and promoted carcinogenesis. AhR maintained stemness of NSCLC cells via JAK2/STAT3 pathway. Ruxolitinib inhibited JAK2/STAT3 signaling and eliminated the tumor sphere forming abilities of AhR-wild-type expressing-PC-9 cells [31].

AhR is actively involved in the metastatic spread of melanoma cells. Significantly, syngeneic C57BL/6J mice transplanted intra-cutaneously with AhR knockout-HCmel12 melanoma cells exhibited a substantial reduction in the frequency and number of pulmonary metastases [32].

IκB kinase α (IKKα) worked synchronously with AhR and transcriptionally upregulated cancer-stemness related gene network consisting of ABCG2, c-Myc, ALDH1A1, Lgr6 and KLF4 [33]. Collectively, the data unveil an interesting role of IKKα in the transcriptional regulation of different genes.

Biseugenol decreased the growth of primary tumors, peritoneal extension and liver, spleen and lung macro-metastases. A gastric cancer model was used for the evaluation of anticancer and anti-metastatic effects of biseugenol. The animal model study also revealed that inhibition/inactivation of AhR or Biseugenol-induced Calpain-10-activation suppressed the growth of tumor mass and reduced the frequency of peritoneally disseminated cancer cells. Biseugenol activated Calpain-10-mediated cleavage of AhR in the cancer cells (Figure 3). AhR is involved in transcriptional upregulation of epithelial-to-mesenchymal (EMT)-related transcriptional factors. However, Calpain-10-induced cleavage of AhR prevented transcriptional upregulation of SNAIL (Figure 3) [34].

AhR promoted pro-survival signaling in breast cancer cells. AhR knockdown resulted in the induction of apoptotic death in breast cancer cells. For experimental metastasis, MDA-MB-231 breast cancer cells were injected into the lateral tail vein of mice. Pulmonary metastatic nodules were lower in number and smaller in size in animal models injected with AhR-silenced MDA-MB-231 cancer cells [35].

There was an evident decrease in the number of metastatic nodules on the surface of lungs from mice injected with TDO2-inhibitor treated MDA-MB-231 cancer cells (Figure 3). TDO2 inhibition severely impaired the capability of triple negative breast cancer cells to effectively metastasize to secondary organs following tail vein injections in mice [36].

Conditioned media from particulate matter-treated macrophages increased cancer cell motility and activated EGFR signaling in cancer cells. HBEGF (heparin-binding EGF-like growth factor) secreted by particulate matter-treated macrophages activated EGFR in cancer cells [37]. Particulate matter exposure led to nuclear accumulation of AhR as well as AhR-mediated upregulation of HBEGF in macrophages. Intratracheal injections of particulate matter were introduced into the lungs. Consequently, particulate matter was deposited in the lung tissues along with an increase in the infiltration rate of inflammatory cells into the bronchioles and alveolar tissues [37]. The total number of cells in the bronchoalveolar lavage fluid was found to be increased in mice injected with particulate matter. The metastatic spread of B16F10 cells to the lungs was increased in mice exposed to the particulate matter. Expression of HBEGF was enhanced in lung tissues. Precisely, it was verified that HBEGF was increased specifically in macrophages isolated from the lungs of experimental mice treated with particulate matter [37].

AhR has also been reported to behave as a critical suppressor of a robust immunological response against tumorigenesis. AhR-silenced-MOC1 murine oral cancer cells were orthotopically implanted into the tongues of C57BL/6J mice [38]. There was a significant retrogression of the tumors generated from AhR-silenced-MOC1 cells by week 2 and sustained inhibition of tumor growth was reported over a time span of 7 weeks. Importantly, the number of CD4+ and CD8+ cells was significantly higher in the tumor-draining lymph nodes in AhR-silenced-MOC1-implanted mice. Remarkably, an increase in the frequency of CD4+ and CD8+ cells triggered rapid clearance of AhR-silenced-MOC1 cells [38]. The percentage and number of CD4+ and CD8+ T cells within tumor-draining lymph nodes returned back to the baseline levels after clearance of the tumors in AhR-silenced-MOC1-implanted mice. Earlier exposure of the mice to AhR-silenced-MOC1 tumors triggered the development of a strong immunological memory against the neoantigens expressed in MOC1-wild type cells. AhR-silenced-MOC1-injected mice re-challenged orthotopically with MOC1-wild type cells after 100 days did not show any evidence related to the development of wild-type tumors [38].

Formate is a metabolite having tumor promoting activities. Formate is produced by colorectal cancer-associated bacterium *Fusobacterium nucleatum* (Fn) [39]. Formate fueled invasive abilities of cancer cells by activation of AhR-induced stemness of cancer cells. Metastatic dissemination of the cancer cells to the lungs was noted to be higher after tail vein injections of T18 cancer cells pre-treated with Fn. However, use of an AhR inhibitor led to a reversal of Fn-induced metastatic dissemination. Intratumoral injections of Fn stimulated the levels of formate within tumor interstitial fluids. Fn exerted pro-inflammatory effects and promoted the expansion of CD4+IL-17+RORγT+ T cells. It is important to note that Th17 cells are a subset of CD4+ T cells classically characterized by a master transcriptional factor RORγt and the production of interleukin-17. Fn promoted the specified expansion of CD4+IL-17+RORγT+ T cells in the lamina propria of experimental models treated with Fn [39].

Particulate matter 2.5 (PM2.5) is a risk factor for the progression of lung cancer. PM2.5 activated AhR, promoted its nuclear translocation and potentiated AhR-mediated transcriptional activation of TMPRSS2 in A549 and H1975 cancer cells. Tumors derived from TMPRSS2-1-depleted H1975 cells were smaller in size in tumor-bearing mice [40].

Binding of interleukin-2 (IL-2) to interleukin-2R (IL-2R) resulted in the recruitment and activation of the Janus kinases. JAK mediated phosphorylation of STAT5-induced exhaustion of CD8+ T cells [41]. IL-2 has the ability to upregulate the expression of Tryptophan hydroxylase 1 (TPH1) in the resting and activated CD8+ T cells. TPH1 catalyzed the conversion of tryptophan to 5-hydroxytryptophan (5-HTP). 5-HTP activated nuclear accumulation of AhR and prompted AhR-mediated upregulation of inhibitory receptors. Blockade of IL-2 cascade caused a decline in AhR levels in the activated CD8+ T cells, which clearly indicated that the IL-2 pathway regulated the activity of AhR. Intratumoral injections of 5-HTP into B16- or MC38-tumor-bearing experimental mice led to the upregulation of inhibitory receptors and simultaneously reduced the production of effector molecules interferon-gamma and tumor necrosis factor in CD8+ TILs [41]. Collectively, these findings clearly suggested that IL-2 transduced the signals intracellularly through the STAT5-5-HTP-AhR pathway and induced exhaustion of CD8+ T cells in tumor microenvironment.

## 3. Tumor Suppressive Roles of AhR

The p-hydroxycinnamic acid inhibited cancer progression partly through the activation of AhR signaling [42].

Glypican-5 (GPC5) is a member of heparan sulfate proteoglycans and reportedly involved in the regulation of cancer. GPC5 regulated the expression of CTDSP1 (C-terminal domain small phosphatase-1) and played a central role in the suppression of cancer progression. AhR was translocated into the nucleus from the cytoplasm in GPC5-overexpressing lung cancer cells. AhR/ARNT heterodimers transcriptionally upregulate CTDSP1. GPC5 overexpression impaired the lymph node metastasis of lung cancer cells in animal models [43].

There was a significant increase in the metastatic spread in mice orthotopically implanted with AhR-silenced H1975 cancer cells in the lungs. AhR stimulated the expression of ATF4 and consequently ATF4 triggered the transcription of ASNS (asparagine synthetase). Asparagine synthetase (ASNS) catalyzed ATP-dependent biosynthesis of L-asparagine from L-aspartic acid and limited the uncontrolled growth of cancer cells [44].

AhR activation caused inhibition of the hypoxia-induced increase in the expression of VEGF in LNCaP and PC-3 cells (Huang). ARNT is a common dimerization partner of HIF-1α and AhR. Therefore, AhR competed with HIF-1α for binding with ARNT and impaired the formation of the prostate cancer by suppression of the production of VEGF. For the analysis of tumor suppressive functions of AhR in prostate carcinogenesis, AhR-null mice were genetically crossed with transgenic adenocarcinoma of the mouse prostate (TRAMP) model of prostate cancer. AhR+/+ TRAMP mice demonstrated markedly reduced tumor development [45].

## 4. Complex Interplay between Ubiquitin Ligases and AhR

In 2004, Ciechanover, Hershko and Rose won the Nobel Prize (Chemistry) in recognition of their ground-breaking discoveries, which unraveled novel mechanisms associated with ubiquitin-directed degradation of substrates [46,47,48,49,50,51]. The fate of the ubiquitinated proteins, however, can be reversed through the action of deubiquitinases [52,53,54,55]. In this section, we have selected key findings for discussion related to regulation of ubiquitin ligases by AhR in different cancers. Moreover, we have also highlighted how AhR interacted with different ubiquitin ligases and deubiquitinating enzymes for regulation of signaling pathways.

Ubiquitin-specific protease 14 (USP14) has gained attention because of its ability to interfere with ubiquitination and degradation of oncogenic proteins [56,57,58,59,60,61,62,63]. USP14, a deubiquitinating enzyme interacted with and stabilized IDO1 in colorectal cancer cells [64]. TRIM21 (Tripartite motif containing-21), a cytosolically located ubiquitin ligase ubiquitinated IDO1 via K48-linkage and promoted degradation (Figure 4A). TRIM21 overexpression caused a significant increase in the ubiquitination of IDO1 [64], whereas ubiquitination of IDO1 was noted to be reduced in TRIM21-depleted cancer cells. USP14 overexpression markedly increased kynurenine levels, while IDO1 knockdown caused a decline in the levels of kynurenine in the USP14-overexpressing tumors. USP14 overexpression reduced the percentage of Granzyme B+ CD8+ T cells, but expanded the proportion of CD25+FOXP3+ CD4+ T cells in tumors (Figure 4B). Conversely, IDO1 knockdown restored the percentage of Granzyme B+ CD8+ T cells and reduced the frequency of CD25+FOXP3+ CD4+ T cells within tumor tissues derived from USP14-overexpressing cancer cells. USP14 inhibition led to a marked reduction in the tumor growth, augmented the infiltration of cytotoxic T cells, reduced the infiltration rate of Regulatory T cells (Tregs) and promoted the response to anti-PD-1 therapy [64]. These findings have described a complex interplay among IDO1, TRIM21 and USP14 in cancer progression. Stability of IDO1 results in an increase in the production of kynurenine; however, the concise role of AhR needs to be investigated comprehensively in the context of IDO1, TRIM21 and USP14. However, circumstantial evidence also sheds light on the off-target effects of IDO1 inhibitors. There is evidence of compensatory activation of AhR pathway in cells treated with IDO1 inhibitors. USP14 inhibition reduces both IDO1 protein levels and IDO1-mediated immunosuppression. Moreover, USP14 inhibition efficiently abolished the off-target effects related to IDO1 inhibitors-mediated activation of AhR.

### 4.1. Oncogenic Role of UCHL3 in Stabilization of AhR and Cancer Progression

Growing evidence demonstrates that high UCHL3 (ubiquitin C-terminal hydrolase L3) activity promotes cancer onset and progression. UCHL3 is a deubiquitinating enzyme having the unique ability to deubiquitinate and extend the half-life of different proteins. Due to their wide-ranging regulatory roles in key processes, UCHL3 might provide new therapeutic targets [65,66,67,68,69,70,71]. UCHL3 is involved in the stabilization of AhR. UCHL3 promoted tumor stem-like properties through stabilization of AhR. It was shown that AhR transcriptionally upregulated ABCG2, c-Myc, ALDH1 and KLF4 in A549 cells. Tumor growth was noted to be significantly impaired in mice inoculated with UCHL3-silenced H358 cells [72].

Importantly, UCHL3 is reported to be negatively modulated by tumor suppressor microRNA. miRNA-582-5p directly targeted UCHL3 and inhibited cancer progression (Figure 4E). However, an oncogenic long non-coding RNA has been described to interfere with miRNA-582-5p-mediated targeting of UCHL3 [73]. Injections of LINC00665-overexpressing PC9 cells effectively promoted the tumor growth in irradiated mice. There was an evident increase in the levels of LINC00665, UCHL3, AhR and PD-L1, but miRNA-582-5p was downregulated in the tumor tissues derived from LINC00665-overexpressing cancer cells. Tumor tissues of irradiated mice xenografted with LINC00665 overexpressing cancer cells displayed elevated levels of PD-1 and PD-L1 and fewer CD8+ T cells [73]. Overall, these findings indicated that LINC00665 enabled NSCLC cells to evade immune elimination via UCHL3-directed stability of AhR protein primarily through miRNA-582-5p/UCHL3 regulatory axis. UCHL3 is an effective pharmacological target to block AhR-driven signaling during cancer progression.

Stability of AhR is imperative for downstream signaling and SUMOylation has been shown to stabilize AhR. AhR has been reported to undergo post-translational modifications by SUMOylation. Two SUMOylation sites have been identified in AhR [74]. One is in the bHLH domain and the other site resides within the TAD domain of AhR. Nuclear fractions of TCDD-treated cancer cells showed a reduced level of SUMOylated AhR. SUMOylation stabilized AhR via inhibition of its ubiquitination-mediated degradation. However, SUMOylation also exerted repressive effects on the transactivation activities of AhR. SENP1 is a crucial deSUMOylating enzyme. There was an evident increase in AhR-mediated transcriptional activity after deSUMOylation [74].

### 4.2. Transcriptional Regulation of Ubiquitin Ligases by AhR

Indole-3-Carbinol is a phytochemical having notable cancer inhibitory properties. Indole-3-Carbinol is also an AhR ligand and triggers AhR-mediated transcriptional upregulation of UBE2L3 (Ubiquitin Conjugating Enzyme E2 L3). UBE2L3 ubiquitinated human papillomavirus (HPV)-encoded-E7 oncoprotein (Figure 4C,D) [75]. These findings are intriguing and warrant comprehensive research. However, UBE2L3 has also been shown to inhibit apoptosis and promote carcinogenesis [76]. AhR transcriptionally upregulated UBE2L3 and directed specific proteins for degradation. UBE2L3 increased ubiquitination and degradation of p53. After TCDD treatment, wild-type mice displayed notable reduction in the levels of p53 levels. However, decline in the levels of p53 was not observed in TCDD-treated AhR-null mice. AhR activation by TCDD induced an increase in the levels of ubiquitinated p53 [76].

RNF182 (Ring Finger Protein-182) is an E3 ubiquitin ligase [77,78]. RNF182 knockdown considerably increased colony formation and proliferation of the cancer cells. Benzo[a]pyrene (BaP), an AhR agonist not only increased nuclear accumulation of AhR, but also transcriptionally downregulated RNF182 in cancer cells [79]. BaP enhanced carcinogenesis by transcriptional repression of RNF182 primarily through promoting the recruitment of AhR to the promoter of RNF182.

AhR deficiency drastically impaired the infiltration of lymphocytes into the tumor microenvironment. AhR ^–/–^ NK cells failed to infiltrate the tumors to a greater extent as compared to the wild-type NK cells [80]. ASB2 (Ankyrin repeat- and SOCS BOX-containing protein-2) encodes a subunit of a multimeric E3 ubiquitin ligase complex. AhR transcriptionally upregulated ASB2 in NK cells. ASB2-deficiency phenocopied AhR-deficiency in natural killer cells in terms of the capability of natural killer cells to invade and infiltrate tumor tissues. Filamin A inhibited the migration of natural killer cells. However, AhR-mediated upregulation of ASB2 led to ubiquitination and degradation of Filamin A [80].

Carboxy-terminus of HSC70-interacting protein (CHIP/STUB1) is a 35 kDa protein having remarkable E3 ubiquitin ligase activity. Additionally, 2-(4-hydroxy-3-methoxyphenyl)-benzothiazole (YL-109) triggered AhR-mediated transcriptional upregulation of CHIP/STUB1. YL-109 impaired metastatic dissemination of MDA-MB-231 cancer cells to the lungs of experimental mice (Figure 4F) [81].

### 4.3. AhR Works with Ubiquitin Ligases for Regulation of Different Proteins

Rbx1, a RING-domain E3 ligase ubiquitinated and degraded ERα. Indole-3-carbinol induced the ubiquitination and proteasome-mediated degradation of ERα [82]. Rbx1 and AhR worked as a complex to trigger the ubiquitination of ERα. Degradation of ERα resulted in transcriptional repression of GATA3. It has been shown that GATA3 transcriptionally upregulated ERα. Therefore, Indole-3-carbinol effectively reduced the levels of ERα by blockade of Rbx1-mediated degradation and GATA3-mediated transcriptional upregulation [82].

The detection of foreign DNA is a crucial alarm for the activation of immunological responses. This is a well-orchestrated response and regulated by cyclic GMP-AMP synthase (cGAS)-stimulator of interferon genes (STING) cascade. Previous studies had shown that the binding of cGAS to dsDNA (double-stranded DNA) caused an allosteric activation of its catalytic activity and resulted in the production of 2’3’ cyclic GMP–AMP (cGAMP). Importantly, cGAMP is not only a second messenger molecule but also a strong agonist of STING. AhR reduces the stability of STING by the formation of a signalosome with ubiquitin ligases [83]. CUL4B forms a complex with RBX1 (RING-box protein-1). AhR promotes the interaction between CUL4B and STING and enhances the addition of K48-linked polyubiquitin chains. CUL4B-knockout cells formed significantly smaller tumors in mice. Moreover, tumor forming ability of double knockout cells (CUL4B and STING) was found to be considerably enhanced [83].

## 5. Moonlighting Activities of AhR as a Ubiquitin Ligase

Carbidopa, a peripheral decarboxylase inhibitor is an AhR agonist. Carbidopa efficiently promoted AhR-mediated AR ubiquitination and proteasomal degradation [84]. Intraperitoneal injections of Carbidopa induced regression of the tumor mass in mice xenografted with LNCaP cancer cells [84].

AhR acted as a ligand-activated E3 ubiquitin ligase and targeted ERα for proteasomal degradation. AHRR (Aryl hydrocarbon receptor repressor) masked the activities of AhR as a transcriptional factor and promoted E3 ubiquitin ligase functions of AhR. The tumor growth rate was found to be significantly impaired in mice inoculated with AhRR-overexpressing-MCF7 cancer cells [85].

Overexpression of AhR induced SMAD4 ubiquitination and proteasomal degradation. Moreover, AhR, JAB1 (Jun-activation domain binding protein) and SMAD4 interacted and formed a complex that induced ubiquitination of SMAD4 in AhR-overexpressing H1299 cancer cells. Overexpression of AhR caused suppression of invasive properties of cancer cells [86].

The AhR-vimentin protein complex is formed in the cytoplasm resulting in proteasomal degradation of vimentin. There was a significant increase in the number of pulmonary metastatic nodules in H1299-wild-type-xenografted mice, whereas a decrease in the number of metastatic colonies was seen in AhR-overexpressing-H1299-bearing mice [87].

Icaritin, a natural prenylflavonoid efficiently inhibited prostate cancer signaling [88]. Icaritin promoted AhR mediated degradation of both androgen receptor and androgen receptor-variants. Intraperitoneal administration of Icaritin impaired the growth of LNCaP tumors in mice orthotopically implanted with androgen-sensitive LNCaP cells into the prostates. Similar results were recorded in mice orthotopically implanted with CWR22Rv1 cancer cells. The size of the tumors derived from CWR22Rv1 cancer cells was smaller in size in mice intraperitoneally administered with Icaritin [88].

## 6. Epigenetics Related to AhR-Mediated Downstream Signaling

Epigenetics is a highly complicated mechanism and phenomenal advancements have been made in the comprehensive characterization of methylation-associated machinery [89,90,91,92,93,94,95,96]. AhR has been shown to utilize different mechanisms to regulate epigenetics. AhR has the ability to modulate the expression of regulators of methylation-associated machinery. Moreover, AhR has also been studied to mediate the process of epigenetics by orchestrating the interaction of long non-coding RNAs with methyltransferases.

In this section, we have provided a summary of different mechanisms that enable us to gain insights into the complicated interaction between AhR and epigenetic machinery in the regulation of cancer.

### 6.1. Regulation of AhR by Epigenetic Machinery

NR2E3, an orphan nuclear receptor, formed a transcriptional complex with SP1 as well as GRIP1 in cancer cells [97]. Findings suggested that a multi-protein complex stimulated the expression of AhR. NR2E3 loss promoted the recruitment of LSD1 (Lysine-specific histone demethylase-1). Consequently, LSD1 reduced the levels of H3K4me2 and transcriptionally repressed AhR (Figure 5). Levels of AhR and H3K4me2 were found to be significantly reduced in the livers of Nr2e3rd7 (Rd7) mice that expressed low levels of NR2E3. Use of LSD1 inhibitors led to an increase in the levels of AhR and H3K4me2 in Rd7 mice. Moreover, AhR knockout mice exhibited a notable increase in the development of diethylnitrosamine-induced liver tumors [97].

Brominated alkaloid Isofistularin-3 (Iso-3), from the marine sponge *Aplysina aerophoba* induced de-methylation of AhR in cancer cells. There was an evident increase in the expression of AhR in Isofistularin-3-treated cancer cells [98]. It was shown that Isofistularin-3 demethylated AhR and induced apoptosis in cancer cells. However, there are visible knowledge gaps in the study. Whether there is any relationship between AhR and apoptosis-inducing ligands (TRAIL) needs to be determined. How AhR regulates apoptosis in TRAIL-treated cancer cells is an exciting area of research.

### 6.2. AhR Mediated Regulation of Cancer-Associated Genes Is Influenced by Epigenetics

Different studies provide evidence that AhR-mediated regulation of different proteins is influenced by epigenetic modifications. DNA methylation interferes with AhR mediated regulation of genes. Sulfotransferase Family 1C member 2 (SULT1C2) is a phase II detoxifying enzyme scientifically acknowledged for its capability to metabolize xenobiotics. The AhR binding site has been identified within critical methylation sites upstream of SULT1C2. Cigarette smoke condensate exposure considerably enhanced the attachment of AhR to the binding sites in the promoter region of SULT1C2 in multiple lung cell lines. Overall, these findings indicated that CSC exposure resulted in the activation of AhR and increased its binding to the promoter region of SULT1C2. AhR triggers the upregulation of SULT1C2, but this process is hampered mainly because of DNA methylation at the promoter region of SULT1C2 [99].

AhR has the ability to trigger the expression of Decorin. However, CpG methylation severely impaired the binding of AhR to the promoter region of Decorin. Demethylating agent 5-Aza significantly induced re-expression of Decorin, reduced the levels of p-SMAD3 and simultaneously increased the levels of E-cadherin in 95D cells [100]. Therefore, AhR stimulated the expression of tumor suppressors and promoted the levels of E-cadherin.

### 6.3. AhR Mediated Regulation of Proteins Played Important Role in Epigenetic Modifications

Studies show that AhR transcriptionally upregulates certain non-coding RNAs (MALAT1) and histone methyltransferases (SUV39H1) and thus regulates epigenetic modifications.

Environmental toxicants have the ability to trigger the stimulation of non-coding RNAs. TCDD (2,3,7,8-tetrachlorodibenzo-p-dioxin) acted as an AhR agonist and efficiently induced MALAT1 in AsPC-1 and PANC-1 cancer cells [101]. AhR transcriptionally upregulated MALAT1. MALAT1 worked synchronously with EZH2 and increased the levels of H3K27me3 (Figure 5). TCDD treatment resulted in a significant rise in the levels of MALAT1, EZH2, and H3K27me3 levels but the cells co-treated with TCDD and AhR antagonists demonstrated marked reduction in the levels of MALAT1, EZH2 as well as H3K27me3 [101]. Collectively, ligand-dependent and environmental toxicants-induced regulatory role of long non-coding RNAs by AhR should be examined comprehensively. Additionally, role of AhR and MALAT1 can be studied in pancreatic cancer model by inoculation of AhR-overexpressing pancreatic cancer cells.

Benzo[a]pyrene (BaP) and arsenic are among the most common environmental pollutants. These chemicals have the ability to enhance lung carcinogenesis. Arsenic and BaP synergistically induced cellular transformation, cancer stem cell-like properties and tumorigenesis [102]. Importantly, Histone-lysine N-methyltransferase (SUV39H1) trimethylates lysine 9 of histone H3. Higher levels of H3K9me2 in arsenic and BaP co-exposure-transformed cells are regulated by SUV39H1. AhR transcriptionally stimulated the expression of SUV39H1 (Figure 5). Co-exposure of cancer cells with arsenic and BaP increased the enrichment of H3K9me2 in the promoter region of SOCS3 as compared to BaP exposure alone (Figure 5). Furthermore, stable knockdown of SUV39H1 not only decreased the levels of H3K9me2, but also stimulated an increase in the levels of SOCS3 in arsenic and BaP co-exposure-transformed cells [102]. These results suggested that AhR demonstrated a remarkable ability to epigenetically inactivate tumor suppressor genes via regulation of methyltransferases.

### 6.4. AhR Worked with Epigenetic-Modifying Proteins

It has previously been revealed that AhR epigenetically inactivated BRCA in breast cancer cells. TCDD activated AhR and promoted its nuclear accumulation. DNA methyltransferases (DNMT1, DNMT3A, DNMT3B) have critical roles in DNA methylation. AhR worked synchronously with DNMT1, DNMT3A, DNMT3B and MBD2 (methyl binding protein-2). There was an evident increase in H3K9me3 at the promoter region of BRCA1, whereas the use of AhR antagonists interfered with crosstalk of AhR with methylation-associated machinery and stimulated the expression of BRCA1 [103,104].

Aza-PBHA, an effective histone deacetylase inhibitor, increased NRF2-mediated upregulation of AhR. Aza-PBHA promoted the formation of AhR-HDAC complexes for the inactivation of HDAC activity [105]. Overall, these findings indicated that HDAC inhibitors impaired the activity of HDAC to impair invasive properties of cancer cells by facilitating an increase in the formation of AhR/HDAC complexes for the inhibition of HDAC activities and stability of histone acetylation patterns.

Hypermethylation blocked the binding of Sp1 to the promoter region of AhR. Use of demethylating agent caused an increase in the levels of AhR in acute lymphoblastic leukemia (ALL) cells [106]. These findings have to be investigated in animal models for a clear role of AhR in acute lymphoblastic leukemia.

Certain clues have highlighted epigenetic-based synergism involving AhR and interleukin-4 in the upregulation of chemokine CCL1 expression in human cells. These findings supported the significance of exposure to environmental PAHs in differential alteration in the functions of macrophage subsets. There is a need to study the interplay between AhR and epigenetic modifications in immune regulatory cells in a detailed way [107].

## 7. Targeted Inhibition of AhR

The main aim of discussion related to AhR targeting is to expand the list of pharmacological drugs for effective targeting of AhR-overexpressing cancers. Combinatorial strategies consisting of epigenetic-modifying drugs and AhR antagonists will be helpful in realistic and evidence-based results for cancer inhibition. Hopefully, these modalities have the potential to offer patients new treatment options across diverse indications. Studies had shown that overexpression and activation of AhR pathway led to significant impairment in the development and functions of NK cells against AML [108].

StemRegenin1 (SR1) is an antagonist of AhR demonstrating unique ability to generate natural killer cells having efficient interferon-γ production capability and cytolytic activity against AML and multiple myeloma cells. These NK cells are generated from CD34+ progenitor cells and display high efficiency. These findings indicate that SR1 generated high fraction of functionally capable NK cells from CD34+ HSPCs providing exceptional promise for clinically efficient NK-cell-based immunotherapy [109,110,111].

AhR inhibition promoted the differentiation of tonsillar IL-22-producing innate lymphoid cell type to IFNγ-producing cytolytic mature NK cells [112].

Design of AhR targeting PROTACS is also an exciting strategy to prevent AhR-overexpressing cancers [113]. Excitingly, the concept of a PROTAC (Proteolysis-targeting chimera) molecule harnessing the ubiquitin-proteasome system for the targeted degradation of an oncogenic protein is very valuable in molecular oncology. Above all, clinical proof-of-concept for PROTAC molecules against two well-established cancer targets in 2020 has substantiated the significance of PROTACs in clinical settings. PROTAC^®^ Protein Degrader ARV-471 has demonstrated notable clinical benefit rate in locally advanced or metastatic ER+/HER2− breast cancer patients.

## 8. Clinical Trials

Highly promising preclinical data related to regulation of AhR-driven signaling by different chemicals have gradually opened new horizons for rationally designed clinical evaluations.

AhR inhibitor (BAY 2416964) is an open-label, Phase 1, first-in-human clinical trial for realistic evaluation of AhR inhibitors in cancer inhibition (NCT04069026). The synergistic effects of BAY 2416964 in combination with Pembrolizumab (FDA-approved PD-1 inhibitor) are also being investigated (NCT04999202). IK-175, an orally administered AhR antagonist, is also currently being evaluated with PD-1 inhibitors (nivolumab) for clinical efficacy (NCT04200963).

These results are informative and future research should be focused on the rationalization of combination therapies in the frontline clinical settings.

## 9. Concluding Remarks

AhR-mediated mechanisms have opened new horizons for an effective cancer therapy. In this review, we have discussed how lessons learnt from clinical trials and an improved understanding of AhR-mediated signaling pathways could galvanize the field. Pharmacological targeting of AhR using different antagonists clearly gives convincing evidence about considerable role of AhR-mediated pathways in cancer progression. Collectively, these findings have underscored the significance of ‘personalized cancer medicine’, an evolving approach to cancer therapy that exploits the increasingly appreciated heterogeneous role of AhR-driven signaling among different subsets of patients.

## Figures and Tables

**Figure 1 cells-12-02382-f001:**
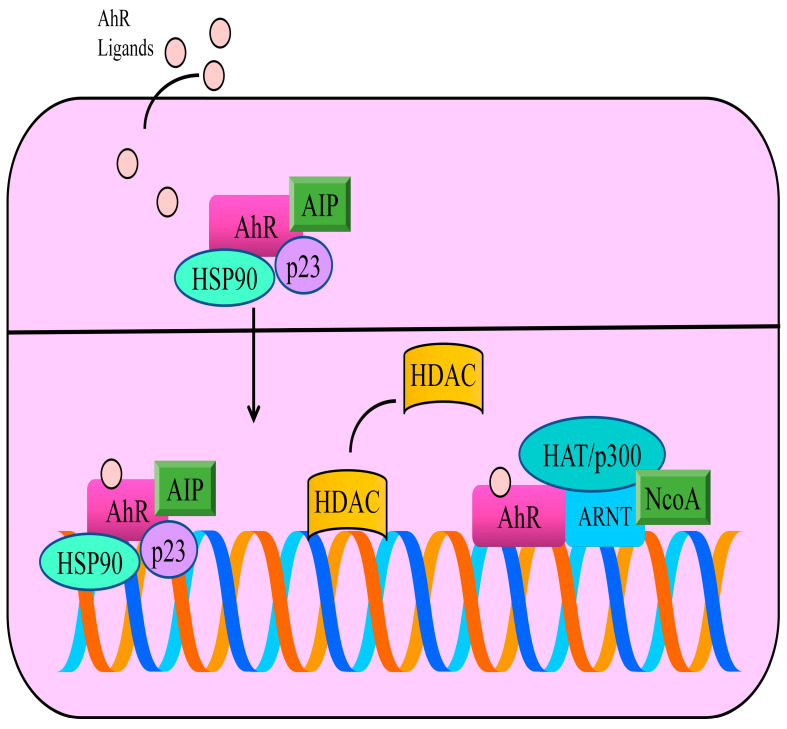
In the inactive state, AhR is primarily located in the cytoplasm and exists as a multi-protein complex with two chaperone proteins, HSP90 (heat shock protein 90) and co-chaperone p23. Moreover, AIP (AhR interacting protein) is a conserved co-chaperone protein that binds to many proteins, including AhR and HSP90. AhR ligands cross the plasma membrane and bind to AhR. These interactions allow the transportation of ligand-receptor complexes into the nucleus. ARNT (AhR nuclear translocator) is a basic Helix-Loop-Helix Motif containing transcriptional factor. Therefore, after accumulation of AhR in the nucleus, AhR forms heterodimers with its partner ARNT. Functionally active heterodimers bind specific DNA regions located in the promoter regions of different genes. AhR works with wide variety of co-factors and chromatin remodeling machinery. The steroid receptor co-activator (SRC) family of p160 proteins consists of SRC-1 (NcoA-1), SRC-2 and SRC-3. AhR/ARNT complex promoted the recruitment of SRC family of transcriptional co-activators. Histone deacetylases (HDACs) are displaced by AhR and histone acetyltransferases are recruited by AhR to stimulate transcriptional gene networks.

**Figure 2 cells-12-02382-f002:**
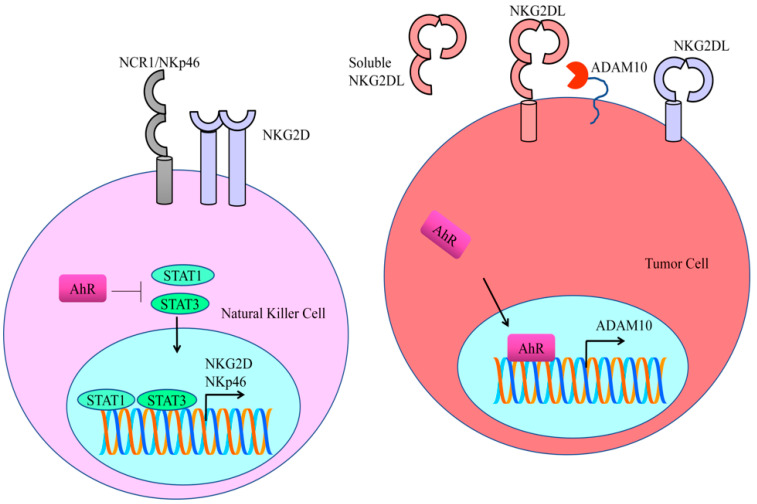
AhR stimulated transcriptional activation of ADAM10. ADAM10-specific inhibitors efficiently reduced the shedding of NKG2DL and enhanced its binding to NKG2D receptors. STAT proteins directly bind to the promoter regions of NK receptors (NKG2D and NKp46). Treatment with STAT inhibitors resulted in a decline in the expression of NKG2D and NKp46.

**Figure 3 cells-12-02382-f003:**
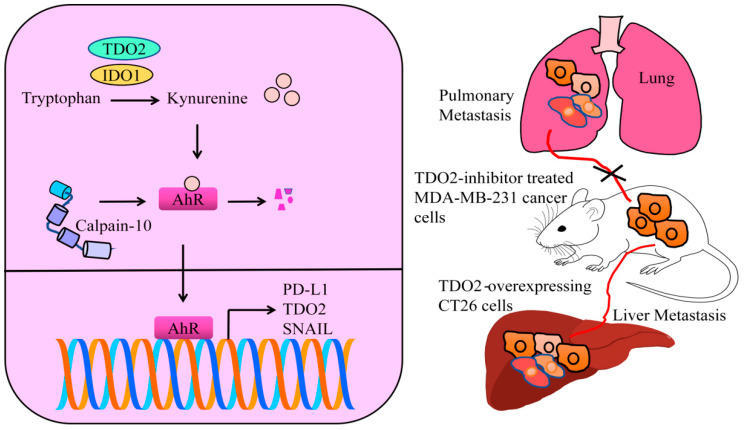
IDO1 and TDO2 catalyze the production of kynurenine. Oncogenic AhR transcriptionally upregulates different oncogenic networks. Calpain-10 has the ability to proteolytically cleave AhR and block AhR-mediated signaling. TDO overexpression promotes metastasis, but inhibition of TDO abolishes metastatic spread.

**Figure 4 cells-12-02382-f004:**
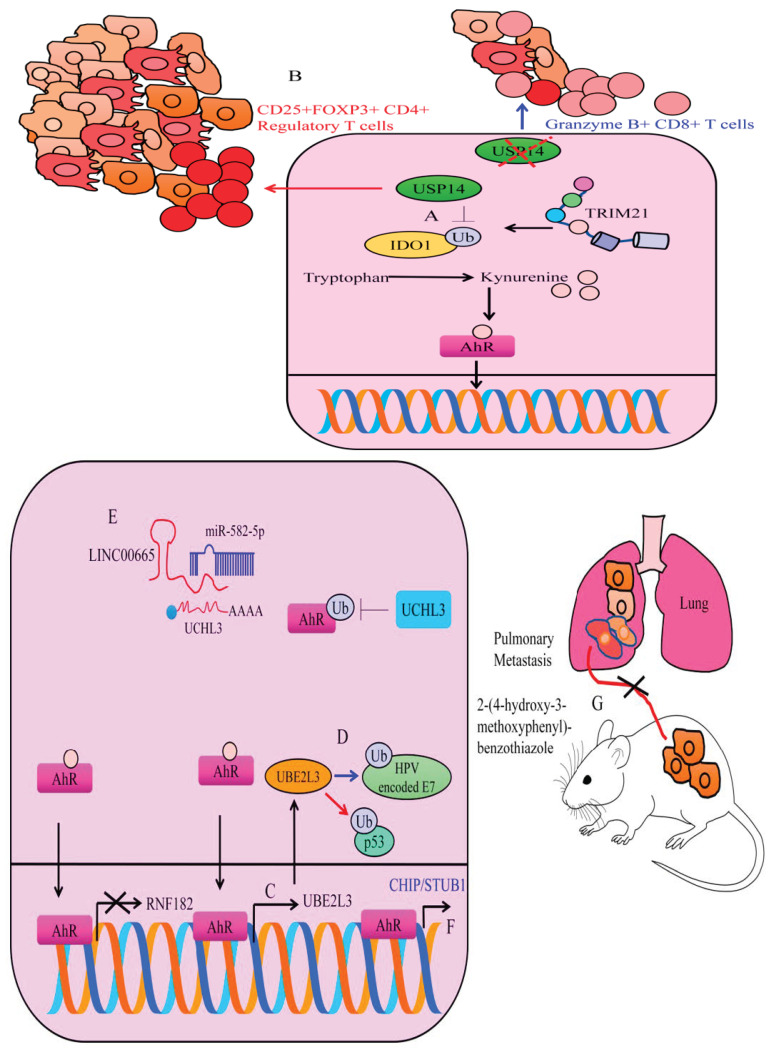
(**A**,**B**) USP14 deubiquitinated IDO1 and enhanced its stability. However, TRIM21 enhanced ubiquitination and degraded IDO1. IDO1 stability caused an increase in the production of kynurenine and activated AhR. Therefore, USP14 promoted tumorigenesis and promoted the mobilization of regulatory CD25+FOXP3+ CD4+ T cells and impaired the immunological response of killer T cells. However, inhibition of IDO1 potentiated the accumulation of Granzyme B+ CD8+ T cells and reduced tumorigenesis. (**C**) AhR transcriptionally regulates different ubiquitin ligases. AhR inactivated RNF182 but enhanced the expression of UBE2L3. (**D**) UBE2L3 has dualistic roles. It not only ubiquitinated and degraded HPV-encoded oncogenic proteins but also tagged tumor suppressor proteins like p53 for degradation. (**E**) UCHL3-mediated cancer promoting effects are inhibited by miRNA-582-5p. However, LINC00665 promoted the expression of UCHL3 and potentiated the deubiquitination of AhR. Stable AhR triggered the expression of oncogenic networks. (**F**) AhR stimulated CHIP/STUB1 and inhibited metastasis. (**G**) YL-109 impaired metastatic colonization of MDA-MB-231 cancer cells to the lungs of experimental mice.

**Figure 5 cells-12-02382-f005:**
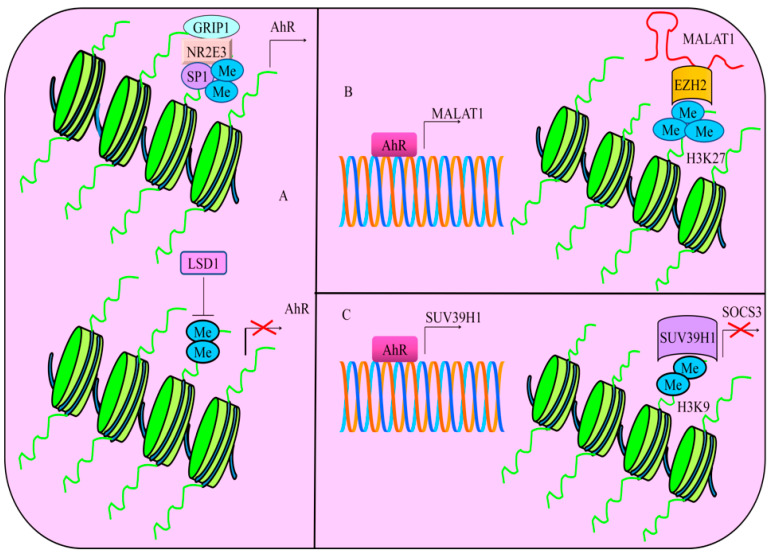
(**A**) Multi-protein complex consisting of NR2E3, SP1 and GRIP1 stimulates the expression of AhR. However, NR2E3 loss promoted the recruitment of LSD1 (Lysine-specific histone demethylase-1). Thus, LSD1 reduces the levels of H3K4me2 and transcriptionally represses AhR. (**B**) AhR transcriptionally upregulated MALAT1. MALAT1 worked synchronously with EZH2 and increased the levels of H3K27me3. (**C**) AhR transcriptionally stimulates the expression of SUV39H1. Resultantly, SUV39H1 transcriptionally represses SOCS3 by increasing the levels of H3K9me2.

## Data Availability

Not applicable.

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
