# Peer review of "Role of Ubiquitination and Epigenetics in the Regulation of AhR Signaling in Carcinogenesis and Metastasis: “Albatross around the Neck” or “Blessing in Disguise”"

_cells, 2023, doi:10.3390/cells12192382_

Round 1

Reviewer 1 Report (Previous Reviewer 1)

The authors followed the comments and suggestions of the reviewer and thus the manuscript has significantly improved. Although some small issues remained or appeared after revision.

Major points:

Lane 131: The new sentence AhR worked with different proteins….: this sentence has no message. Remove it or give it a message

Lane 337: 4.3 AhR works with Ubiquiting ligases:               revise upper case and lower case and spelling, the term is ubiquitin ligase

Lane 418: 6.2 Ahr mediated regulation of Gene Network….: which gene network (s) are meant : again revise upper case and lower case and spelling

Same for the new headings in lane 435: What are target networks?

And lane 460: here: the sentence AhR worked with Epigenetic –Modifying machinery is not really meaningful

Figures: The figures greatly improved, but AhR cofactors are still missing.

Author Response

Major points:

Lane 131: The new sentence AhR worked with different proteins….: this sentence has no message. Remove it or give it a message

We have removed the sentence. 

Lane 337: 4.3 AhR works with Ubiquiting ligases:               revise upper case and lower case and spelling, the term is ubiquitin ligase

We have edited the sentence and corrected the spellings. 

Lane 418: 6.2 Ahr mediated regulation of Gene Network….: which gene network (s) are meant : again revise upper case and lower case and spelling

We have restructured the sentence. 

Same for the new headings in lane 435: What are target networks?

Target networks are gene networks transcriptionally upregulated or downregulated by AhR.

And lane 460: here: the sentence AhR worked with Epigenetic –Modifying machinery is not really meaningful

We have restructured the sentence as "Epigenetic-modifying proteins"

Figures: The figures greatly improved, but AhR cofactors are still missing.

The figure is re-designed with transcriptional co-factors. 

Reviewer 2 Report (Previous Reviewer 2)

All the prior comments are adequately addressed. 

Author Response

We have added a diagram for the EPIGENETICS section. 

This manuscript is a resubmission of an earlier submission. The following is a list of the peer review reports and author responses from that submission.

Round 1

Reviewer 1 Report

The manuscript of Farooqi et al. summarizes “the role of ubiquitination and epigenetics in the regulation of AhR Signaling in Carcinogenesis and Metastasis”. The aspect of epigenetic regulation and the function of ubiquitination processes in AhR signaling is very important and not well understood in the moment. Thus, a review dealing with this subject is very important in the field.

This manuscript, however, leaves many questions unanswered and a common thread is missing.

In part two (page2, oncogenic role of the AhR) the authors start with AhR and IDO, then they switch to NKG2D mediated NK cell activation and then back to AhR and IDO. Especially the term IDO/TDO appears in several parts of this chapter. This should be better structured.

Also several tumor models are listed, but the nature of the models is not explained:

e.g. lane 116: this is a melanoma model, this is not explained

e.g. lane 121: here a model of a gastric tumor is used, this is also not explained nor mentioned

In addition, it is a merely a list of models, there is no summary of the data sets done. There is no relationship or between the data. This also holds true for the rest of the article.

e.g. in the last part, where is comes to ubiquitination: there is a big part about IDO lane 232-244. IDO has a relationship to AhR. Why do the authors not incorporate AhR here? The next sentence starts with UCHL1 activity without transition; this again is a mere list of facts, without connection or context.

Also page 10 targeted inhibition of AhR:

It is interesting, that inhibition of AhR increases NK cell differentiation and development that enhances tumor immunology against AML. But stemregenin is an AhR antagonist, has nothing to do with epigenetics or ubiquitination. The last sentence the PROTACS approach would describe such an item and needs to be further elucidated and discussed.

In a review article the current findings should be summarized, discussed, and agreements or disagreements of the data sets should be discussed. Overall, data or items are listed, without commentary, clues or a roundup of the data.

Further, the manuscript should be revised in English language. In the beginning and in the end it is rather lengthy and imprecisely written, in the middle parts the English language needs improvement.

Figures: the binding partner of AhR, arnt is missing in the figures

Figure 3: in the pink circle a cell as suggested in Figure 1 and 2; why are then cells in the cell?

The manuscript of Farooqi et al. summarizes “the role of ubiquitination and epigenetics in the regulation of AhR Signaling in Carcinogenesis and Metastasis”. The aspect of epigenetic regulation and the function of ubiquitination processes in AhR signaling is very important and not well understood in the moment. Thus, a review dealing with this subject is very important in the field.

This manuscript, however, leaves many questions unanswered and a common thread is missing.

In part two (page2, oncogenic role of the AhR) the authors start with AhR and IDO, then they switch to NKG2D mediated NK cell activation and then back to AhR and IDO. Especially the term IDO/TDO appears in several parts of this chapter. This should be better structured.

Also several tumor models are listed, but the nature of the models is not explained:

e.g. lane 116: this is a melanoma model, this is not explained

e.g. lane 121: here a model of a gastric tumor is used, this is also not explained nor mentioned

In addition, it is a merely a list of models, there is no summary of the data sets done. There is no relationship or between the data. This also holds true for the rest of the article.

e.g. in the last part, where is comes to ubiquitination: there is a big part about IDO lane 232-244. IDO has a relationship to AhR. Why do the authors not incorporate AhR here? The next sentence starts with UCHL1 activity without transition; this again is a mere list of facts, without connection or context.

Also page 10 targeted inhibition of AhR:

It is interesting, that inhibition of AhR increases NK cell differentiation and development that enhances tumor immunology against AML. But stemregenin is an AhR antagonist, has nothing to do with epigenetics or ubiquitination. The last sentence the PROTACS approach would describe such an item and needs to be further elucidated and discussed.

In a review article the current findings should be summarized, discussed, and agreements or disagreements of the data sets should be discussed. Overall, data or items are listed, without commentary, clues or a roundup of the data.

Further, the manuscript should be revised in English language. In the beginning and in the end it is rather lengthy and imprecisely written, in the middle parts the English language needs improvement.

Figures: the binding partner of AhR, arnt is missing in the figures

Figure 3: in the pink circle a cell as suggested in Figure 1 and 2; why are then cells in the cell?

Reviewer 2 Report

The review article by Farooqi et al describes role of AhR in carcinogenesis, ubiquitination, and finally epigenetic studies involving AhR. The article covers all the aspects of latest literature and includes depictions to explain few of the mechanisms. Major Comments:

1)      The way article is written does not integrate impact/role of ubiquitination and epigenetic mechanisms in AhR-mediated progression or attenuation of carcinogenesis. Rather the article is reads very fragmented. A cohesive approach integrating role of ubiquitination and epigenetics into two sections namely tumor progression and tumor suppression will provide a clear and focused message to the readers.

2)      Two illustrations depicting proposed mechanism(s) by which AhR promotes and attenuates carcinogenesis (and role of ubiquitination and epigenetic mechanism) will enhance the quality of the review.